# A voice-based algorithm can predict type 2 diabetes status in USA adults: Findings from the Colive Voice study

Abir Elbéji[1], Mégane Pizzimenti[1], Gloria Aguayo[1], Aurélie Fischer[1], Hanin Ayadi[1], Franck Mauvais-Jarvis[2,3], Jean-Pierre Riveline[4,5], Vladimir Despotovic[6], Guy Fagherazzi[1] *

**1** Deep Digital Phenotyping Research Unit. Department of Precision Health, Luxembourg Institute of Health, 1 A-B rue Thomas Edison, L-1445 Strassen, Luxembourg, **2** Section of Endocrinology and Metabolism, Deming Department of Medicine, Tulane University School of Medicine, New Orleans, Louisiana, United States of America, **3** Southeast Louisiana, VA Medical Center, New Orleans, Louisiana, United States of America, **4** Institut Necker Enfants Malades, Inserm U1151, CNRS UMR 8253, Immediab Laboratory, Paris, France, **5** Department of Diabetology, Endocrinology and Nutrition, Assistance Publique—Hôpitaux de Paris, Lariboisière University Hospital, Paris, France, and INSERM UMR-S1151, CNRS UMR-S8253, Immediab Lab,Institut Necker-Enfants Malades, Université Paris Cité, Paris, France, **6** Bioinformatics Platform, Luxembourg Institute of Health, 1 A-B rue Thomas Edison, L-1445 Strassen, Luxembourg

* guy.fagherazzi@lih.lu

**Data Availability Statement:** Audio data and source codes used in this study are publicly

## Abstract

The pressing need to reduce undiagnosed type 2 diabetes (T2D) globally calls for innovative screening approaches. This study investigates the potential of using a voice-based algorithm to predict T2D status in adults, as the first step towards developing a non-invasive and scalable screening method. We analyzed pre-specified text recordings from 607 US participants from the Colive Voice study registered on ClinicalTrials.gov (NCT04848623). Using hybrid BYOL-S/CvT embeddings, we constructed gender-specific algorithms to predict T2D status, evaluated through cross-validation based on accuracy, specificity, sensitivity, and Area Under the Curve (AUC). The best models were stratified by key factors such as age, BMI, and hypertension, and compared to the American Diabetes Association (ADA) score for T2D risk assessment using Bland-Altman analysis. The voice-based algorithms demonstrated good predictive capacity (AUC = 75% for males, 71% for females), correctly predicting 71% of male and 66% of female T2D cases. Performance improved in females aged 60 years or older (AUC = 74%) and individuals with hypertension (AUC = 75%), with an overall agreement above 93% with the ADA risk score. Our findings suggest that voice-based algorithms could serve as a more accessible, cost-effective, and noninvasive screening tool for T2D. While these results are promising, further validation is needed, particularly for early-stage T2D cases and more diverse populations.

## Author summary

Type 2 diabetes (T2D) is a major public health issue, affecting millions worldwide and leading to severe health complications if undiagnosed. Currently, diagnosing T2D relies on blood tests, which are invasive, costly, and challenging to implement on a large scale.

available in a Github repository. https://github.com/
LIHVOICE/Voice-and-diabetes-VOCADIAB.

**Funding:** Colive Voice study is funded by the
Luxembourg Institute of Health. The French-
speaking Diabetes Society, the Luxembourg
Diabetes Society and the Luxembourg Diabetes
Association further supported this work. The
funders had no role in study design, data collection
and analysis, decision to publish, or preparation of
the manuscript.

**Competing interests:** The authors have declared
that no competing interests exist.

This study explores a new, non-invasive approach: detecting T2D risk through voice anal-
ysis. Using data from the Colive Voice study, we developed a voice-based algorithm to
predict T2D status in adults in the USA. The algorithm analyzes specific voice features
and is designed to capture subtle differences in the voices of individuals with T2D com-
pared to those without. We trained and tested the algorithm separately for men and
women and observed promising results, with the algorithm showing accuracy levels com-
parable to traditional risk assessment tools, such as the American Diabetes Association
(ADA) score. We also found that the algorithms performed better in certain subgroups,
such as older women and individuals with hypertension. Our findings highlight the
potential of voice analysis as an accessible and affordable screening tool for T2D, espe-
cially valuable for early detection in diverse populations and settings with limited
resources. This innovative approach could transform diabetes screening by offering a
practical, scalable solution for identifying those at risk.

## Introduction

Diabetes mellitus (DM) is an endocrine system illness in which the body cannot regulate blood
glucose levels. It is one of the most severe and common chronic diseases of our time, as it was
responsible for 6.7 million deaths in 2021 [1]. In 2022, about 1 in 10 people in the world is liv-
ing with DM, and the number is expected to grow from 537 million adults, up to 643 million
by 2030 and 783 million by 2045, as the result of population aging, economic development,
urbanization, unhealthy eating habits, and sedentary lifestyle[1]. In the USA, according to the
2022 National Diabetes Statistics Report from the CDC [1,2], 37.3 million people, or 11.3% of
the population, have diabetes. This total includes 28.7 million diagnosed cases and an esti-
mated 8.5 million people who are living with undiagnosed diabetes.

One of the most urgent public health challenges in DM is reducing the number of undiag-
nosed cases worldwide. Currently, almost one in every two people with type 2 diabetes (T2D)
is undiagnosed worldwide, and as a result, cannot begin treatment or preventive measures to
avoid or delay complications [3]. It was demonstrated that undiagnosed DM is associated with
a higher death risk when compared to normoglycemic individuals [4], as one-third of T2D
patients do not present symptoms until complications appear [5]. From a health economics
perspective, it has been previously reported that any undiagnosed diabetes case costs $4,250
per year in the USA [6], generating preventable healthcare expenditures.

Nowadays, screening campaigns rely on invasive blood glucose analysis that costs around
825 billion dollars per year [7], which might be difficult to deploy at a large scale or to imple-
ment in countries or settings with limited resources and/or infrastructures. Alternative meth-
ods include scores to identify individuals at risk of developing diabetes during the next 5 to 10
years. The FINDRISC score [7,8] is widely used, although it is based on a questionnaire with
limited detection capacities (AUC around 76%) and can be prone to errors or desirability
biases.

In the United States (USA), The American Diabetes Association (ADA) diabetes risk test[9]
was developed as a screening tool to classify high-risk subjects in the community and to raise
awareness of modifiable risk factors and healthy lifestyles (5). The ADA diabetes risk test scor-
ing includes seven questions (total score of 0–11) regarding age, gender, gestational diabetes
mellitus (GDM), family history of diabetes, high blood pressure, physical activity, and obesity
(based on body mass index (BMI) via a weight-height chart). Those having scores of 5 and
more are considered to be at high risk of having diabetes.

With the advancement of digital technologies and artificial intelligence, significant effort is being directed towards detecting diabetes through noninvasive methods. These methods range from human facial block color analysis using sparse representation classifiers [10], hair analysis through elemental composition [11,12,13], specialized eye exams aimed at detecting diabetic retinopathy [14], to voice analysis, which stands as one of the most promising technologies in healthcare applications. This includes early diagnosis of neurodegenerative diseases [15] and assisting in screening and monitoring symptoms of conditions like COVID-19 [16] through the analysis of subtle speech pattern alterations and vocal biomarkers.

Previous works have suggested that people with diabetes have different voice features than people without diabetes. People with T2D with poor glycemic control or with neuropathy are also more likely to have phonatory symptoms compared to controls [17], namely a higher average score for vocal grading, straining [18], and hoarseness [19] that are affecting patients' quality of life. From an acoustic perspective, it has been shown that voice parameters like jitter, shimmer, smoothed amplitude perturbation quotient, noise-to-harmonic ratio, relative average perturbation, mean fundamental frequencies, maximum phonation time, and amplitude perturbation quotient show significant differences in their values between T2D patients and people without diabetes [20,21]. However, previous studies relied on relatively small sample sizes, a lack of diversity in the participant profiles, and a lack of validation with audio recordings captured in real-world settings.

Building on this groundwork, our study distinguishes itself by leveraging data from the Colive Voice program to develop and assess the performance of a voice-based AI algorithm for T2D status detection in the adult population in the USA. This initiative not only serves as a first step toward using voice analysis as a first-line T2D screening strategy but also offers insights into the complex nature of T2D and its interaction with voice characteristics. Accordingly, we place special emphasis on considering a wide array of demographic and health-related parameters. This holistic approach is crucial as these factors can significantly affect voice characteristics and, consequently, their potential as indicators for disease states.

## Methods

### Study population

In 2021, the Luxembourg Institute of Health initiated a worldwide, multilingual research program named Colive Voice. Its ongoing project serves as a screening platform for vocal biomarkers, for screening or monitoring various chronic diseases and frequent health symptoms. To ensure diversity, Colive Voice collects voice recordings from participants above the age of 15 years, regardless of their health status and conditions, in English, French, German, and Spanish globally. Each participant contributes with standardized vocal tasks which are then annotated with clinical and demographic data.

### Ethics statement

Colive Voice is registered on ClinicalTrials.gov (NCT04848623) and was approved by the National Research Ethics Committee of Luxembourg (study number 202103/01) in March 2021. All participants provided informed consent to take part in the study.

### Collected data

Colive Voice participants are invited to complete a comprehensive questionnaire to gather a diverse range of information: demographic characteristics, lifestyle habits, anthropometric data, symptoms, drug use, and history of chronic diseases. Regarding diabetes, Colive Voice

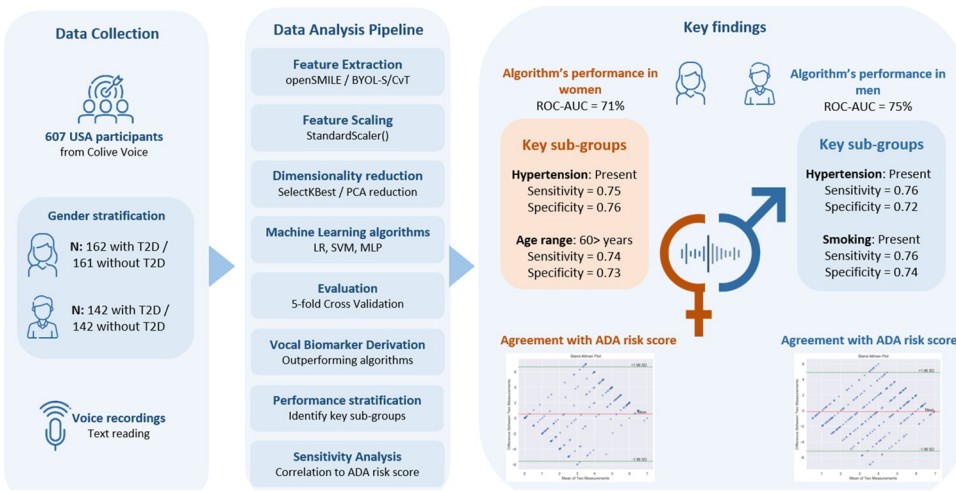

**Fig 1. General workflow.**

gathers data on the diagnosis, type of diabetes, duration since diagnosis as well as treatment categories, and HbA1c levels. For the present work, we included English-speaking participants from the USA and we analyzed each gender separately. Participants were invited to record a standardized reading task using the 25th article from the Human Rights Declaration (Fig 1). All the collected raw audio data was processed and quality-checked to ensure consistency throughout the study. There was no missing data in this study, ensuring a robust and complete dataset for analysis.

## Voice feature extraction

**OpenSmile.** OpenSmile [22] is an open-source toolkit, popularly used for generating handcrafted low-level descriptors (LLD) from audio inputs. These descriptors encapsulate key characteristics of audio signals over time such as pitch, intensity, and spectral properties. OpenSmile computes functionals on these LLD contours, capturing statistical attributes like peaks, means, and ranges to provide a higher-level overview of the audio signal. Among the feature sets that OpenSmile offers, the ComParE set stands out. Comprising 6373 static features, ComParE is notable for its comprehensive nature, offering a rich and extensive array of data points. This vast collection of features facilitates the detection of complex patterns in the audio data, offering an in-depth understanding of the audio source.

## BYOL-S/CvT

The hybrid model, BYOL-S/CvT [23], is a new method that detects cognitive and physical load in speech. It uses both data-driven features from the self-supervised BYOL-S model trained on Audioset and handcrafted features from OpenSmile. This mix improves the model's performance and helps it learn speech patterns better than traditional methods. The BYOL-S/CvT model is also efficient and fast, needing only a single step during the decision-making stage, and produces 2048-dimensional embeddings.

## Data analysis

In this study, the authors adhered to the TRIPOD criteria (the Transparent Reporting of a Multivariable Prediction Model of Individual Prognosis Or Diagnosis) standards for the

reporting of AI-based algorithm development and validation and used the corresponding checklist to guide the drafting of the manuscript.

To mitigate gender bias and manage imbalanced data challenges in our machine learning algorithm training, we first stratified the dataset based on gender. Following this, we used a simple random sampling technique to generate balanced group sizes, ensuring a more equitable and effective training process. Individuals without endocrine diseases, including diabetes, were selected randomly from the general USA population to create a control group that matched the size of the group of participants with T2D.

To enhance our algorithm's performance, we first normalized the extracted features and embeddings using a standard scaler, which helps ensure consistent variance across all features. As high-dimensional inputs could lead to overfitting and poor generalization in machine learning algorithms, we used Principal Component Analysis (PCA) to reduce the dimensionality of the BYOL-S/CvT embeddings. For OpenSmile features, we used feature selection using the SelectKBest function from scikit-learn.

Once the normalization, reduction, and feature selection processes were complete, the resulting features were fed into three different classifiers: Logistic Regression (LR), Support Vector Machine with a radial basis function kernel (SVM RBF), and Multi-Layer Perceptron classifiers (MLP).

To evaluate the performance and compare the classifiers, we used stratified 5-fold cross-validation, ensuring no data leakage via the Pipeline functionality from scikit-learn. This pipeline handled scaling and PCA reduction for the BYOL-S embeddings, as well as scaling and feature selection for OpenSmile features. We measured the algorithms' performances using accuracy, specificity, sensitivity, and AUC metrics.

For optimal results, we fine-tuned the number of PCA components and the algorithms' hyperparameters using the grid search function from scikit-learn. We then used the best feature-classifier combination to select the most performant algorithm for each gender.

## Influence of cofactors and their impact on algorithms' performance metrics

In order to highlight how different cofactors influence the efficacy of our predictive algorithms, we conducted a performance stratification analysis. This analysis was segmented by age (below and over 60 years), BMI (below and over 25–29.9 kg/m$^2$). Additionally, we examined conditions including hypertension, migraine, diagnosed depression, smoking, stress, and fatigue (measured by the Fatigue Severity Scale [24]), designating each condition's status as either 'present' and 'absent' or 'severe' and 'mild'. To reinforce confidence in our performance metrics and facilitate comparisons, we employed a bootstrapping technique. This involved generating multiple subsamples for each combination of comorbidity and its status. The bootstrapping process, repeated for each comorbidity, involves sampling with replacement from the original dataset and subsequently recalculating the metrics for each subsample.

With the objective of developing a screening tool in mind, the assessment of specificity and sensitivity metrics was prioritized, but AUC was also reported. High sensitivity guarantees that true cases are not missed, while high specificity reduces false alarms, optimizing resource use and building user trust. Performance metrics were computed independently for each bootstrap iteration within the respective groups. To evaluate the statistical significance of performance differences between categories, we employed the Mann-Whitney U test. Finally, to account for multiple comparisons, we adjusted the p-values obtained from these statistical tests using a Bonferroni correction.

## Sensitivity analysis

As a sensitivity analysis, we conducted a Bland-Altman analysis between the voice-based algorithms and the ADA risk score, which serves as a gold standard for assessing T2D risk in the USA population [9]. Due to data limitations, physical activity levels and family history of diabetes were not available in Colive Voice and were set to zero for all participants by default. In this context, the modified ADA risk score ranges from zero, denoting no T2D risk, to seven, indicating a high risk.

# Results

## Population characteristics

We analyzed 323 females and 284 males based on T2D status. The majority were identified as white: 73.3% of females with T2D, 76.5% of females without T2D, 77.5% of males with T2D, and 71.8% of males without T2D.

Significant differences were identified across the groups, including age, BMI (t-test p-value < 0.001), and prevalence of hypertension (chi2 p-value < 0.001). Those with T2D, in both genders, had higher average ages and BMIs than those without T2D. Specifically, females with T2D had an average age of 49.5 years and a BMI of 35.8 kg/m$^2$, compared to 40.0 years and 28.0 kg/m$^2$ in those without T2D. Male participants with T2D had an average age of 47.6 years and BMI of 32.8 kg/m$^2$, whereas those without averaged 41.6 years and 26.6 kg/m$^2$.

Hypertension was more prevalent among the T2D group. Among females with T2D, 50% reported hypertension, compared to 11.18% in the group without T2D. For males, a similar trend was observed, with 58.5% of those with T2D having hypertension, compared to 12.7% without the condition.

Depression diagnosis history also was more prevalent in those with T2D (chi2 p-value < 0.001), especially in females: 61.7% with T2D reported depression, compared to 45.3% without T2D. Among males, the rates were 48.6% for those with T2D and 31.7% for those without T2D. Other health conditions and scores are included in Table 1.

## Algorithms' performances

In both genders, MLP classifiers trained with BYOL-S/CvT embeddings significantly outperformed those trained solely on OpenSMILE features in both males and females (Table 2).

For the prediction of T2D in females, the classifier achieved a sensitivity of 0.67±0.11, specificity of 0.66±0.04, an AUC of 0.71±0.07 and a Brier score of 0.31. For the prediction of T2D in males, the reported performance metrics were a sensitivity of 0.73±0.03, specificity of 0.70 ±0.02, an AUC of 0.75±0.05 and a Brier score of 0.22 (Fig 2). The predicted probability of having T2D is then used for the sensitivity analysis with ADA risk score.

## Performance stratification

The specificity and sensitivity metrics showed variability across various subgroups.

When stratifying by key demographics, notable differences were observed for females across age categories, with females aged 60 and above exhibiting higher specificity (0.74±0.12), sensitivity (0.74±0.07), and AUC (0.74±0.07) compared to females aged below 60 for both specificity and sensitivity (0.65±0.04), and for AUC (0.65±0.03) (Table 3).

This table provides an overview of various metrics, differentiated by gender across different demographic factors, comorbidities, and lifestyle factors. The statistical significance of performance differences between categories was evaluated using the Mann-Whitney U test, with all results being statistically significant (p < 0.001).

**Table 1. Study population characteristics.**

| T2D status | Female group | | | Male group | | |
|---|---|---|---|---|---|---|
| | Without T2D | With T2D | P-value | Without T2D | With T2D | P-value |
| Participants (N) | 161 | 162 | - | 142 | 142 | - |
| Age (year) | 40.0 (13.5) | 49.5 (12.1) | <0.001 | 41.6 (14.0) | 47.6 (13.4) | <0.001 |
| Body Mass Index (kg/m$^2$) | 28.0 (7.3) | 35.8 (8.9) | <0.001 | 26.6 (5.5) | 32.8 (8.5) | <0.001 |
| Ethnicity: White | 118 (73.3%) | 124 (76.5%) | 0.28 | 110 (77.5%) | 102 (71.8%) | 0.59 |
| Ethnicity: Black | 20 (12.4%) | 21 (13.0%) | | 10 (7.0%) | 12 (8.5%) | |
| Ethnicity: Other | 23 (14.3%) | 17 (10.5%) | | 22 (15.5%) | 28 (19.7%) | |
| Fatigue Severity Scale | 32.3 (13.4) | 40.3 (12.3) | <0.001 | 31.3 (12.8) | 40.3 (12.3) | <0.001 |
| Perceived stress (% yes) | 38 (23.6%) | 49 (30.3%) | 0.48 | 29 (20.4%) | 38 (26.7%) | 0.16 |
| Smoking (% yes) | 28 (17.4%) | 19 (11.7%) | 0.22 | 32 (22.5%) | 34 (23.9%) | 0.24 |
| Migraine (% yes) | 33 (20.5%) | 43 (26.5%) | 0.25 | 16 (11.3%) | 19 (13.4%) | 0.72 |
| Thyroidic disease (% yes) | 0 (0%) | 37 (22.8%) | <0.001 | 0 (0%) | 10 (0.7%) | <0.01 |
| Hypertension (% yes) | 18 (11.2%) | 81 (50.0%) | <0.001 | 18 (12.7%) | 83 (58.5%) | <0.001 |
| Diagnosed depression (% yes) | 73 (45.3%) | 100 (61.7%) | <0.01 | 45 (31.7%) | 69 (48.6%) | <0.01 |
| HbA1c (%) | - | 7.14 (1.8) | - | - | 7.20 (1.7) | - |
| Diabetes treatment (% yes) | - | 126 (77.8%) | - | - | 114 (80.3%) | - |
| Diabetes duration (year) | - | 8.9 (7.3) | - | - | 9.1 (7.6) | - |

The table presents clinical data describing the overall population of the study. Categorical data are represented by total numbers and percentages, with the calculated p-values derived from chi-square tests. Continuous data are represented by mean and standard deviation, with p-values calculated using the Student's t-test.

Conversely, no noticeable disparities were observed among males.

When considering comorbidities, hypertension emerged as a significant enhancer of the algorithm's performance in both genders. The presence of hypertension enhanced the sensitivity (0.75±0.05 for females and 0.76±0.05 for males), highlighting the algorithm's efficiency in detecting T2D in individuals with hypertension. On the other hand, for females, migraine considerably increases specificity to 0.86±0.07 and sensitivity to 0.75±0.07, while for males with migraine, both specificity (0.67±0.12) sensitivity (0.71±0.11) is lower. This suggests that

**Table 2. Results of the prediction models.**

| | Features | Dimensionality reduction | Classifier | Accuracy | Specificity | Sensitivity | AUC |
|---|---|---|---|---|---|---|---|
| Female group | Opensmile ComParE 2016 (6373) | 200 selected features | LR | 0.60 (0.03) | 0.60 (0.03) | 0.62 (0.07) | 0.62 (0.02) |
| | | | MLP Classifier | 0.63 (0.02) | 0.61 (0.02) | 0.74 (0.02) | 0.66 (0.02) |
| | | | SVM RBF | 0.57 (0.02) | 0.57 (0.02) | 0.63 (0.03) | 0.61 (0.01) |
| | Byol-S embeddings (2048) | PCA, n_components = n_samples | LR | 0.67 (0.04) | 0.68 (0.04) | 0.65 (0.11) | 0.70 (0.06) |
| | | | **MLP Classifier** | **0.67 (0.04)** | **0.66 (0.04)** | **0.67 (0.11)** | **0.71 (0.07)** |
| | | | SVM RBF | 0.66 (0.04) | 0.65 (0.07) | 0.67 (0.11) | 0.71 (0.05) |
| Male group | Opensmile ComParE 2016 (6373) | 100 selected features | LR | 0.56 (0.02) | 0.55 (0.01) | 0.58 (0.05) | 0.61 (0.05) |
| | | | MLP Classifier | 0.61 (0.05) | 0.61 (0.06) | 0.63 (0.06) | 0.64 (0.05) |
| | | | SVM RBF | 0.57 (0.05) | 0.57 (0.06) | 0.54 (0.05) | 0.57 (0.05) |
| | Byol-S embeddings (2048) | PCA, n_components = 100 | LR | 0.69 (0.04) | 0.66 (0.07) | 0.72 (0.03) | 0.73 (0.06) |
| | | | **MLP Classifier** | **0.71 (0.02)** | **0.70 (0.02)** | **0.73 (0.03)** | **0.75 (0.05)** |
| | | | SVM RBF | 0.70 (0.04) | 0.64 (0.05) | 0.76 (0.03) | 0.78 (0.05) |

Table 2 presents the mean and standard deviation (in parentheses) of the performance metrics across cross-validation folds. The selected algorithm for each gender group is highlighted in bold. Logistic Regression (LR), Multi-layer Perceptron (MLP), Support Vector Machine Radial basis function kernel (SVM RBF).

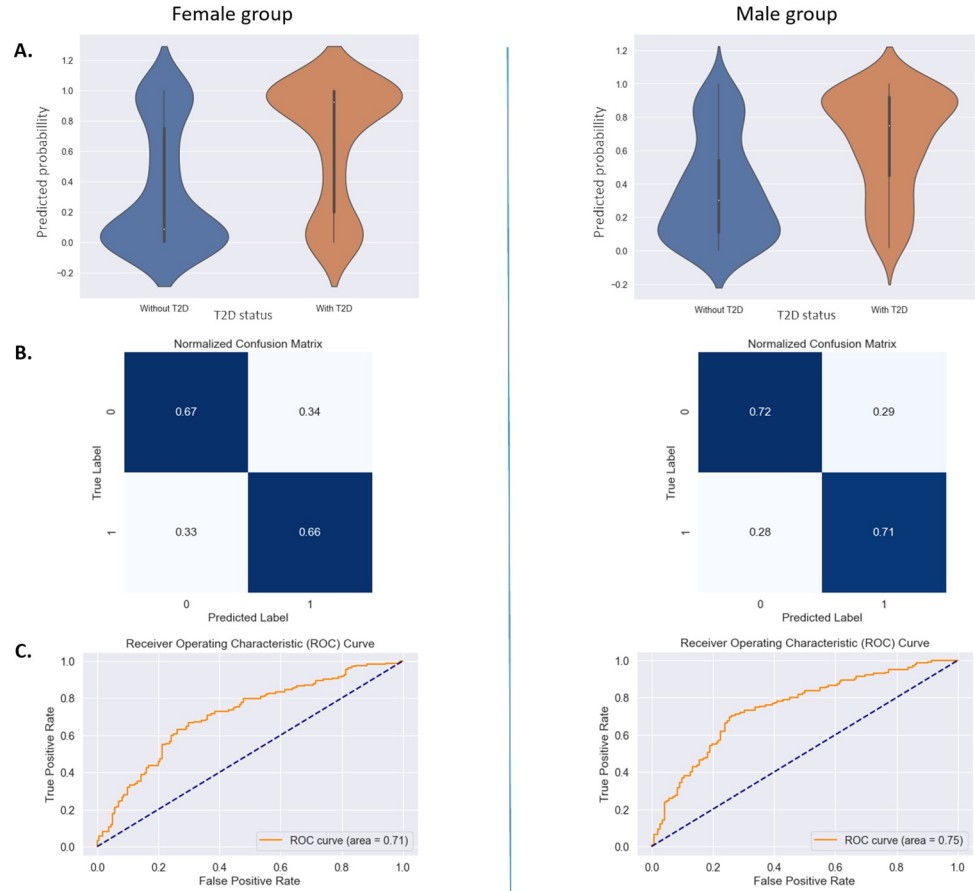

**Fig 2. Voice-based T2D status detection algorithms' overall performance.** A: Predicted probability distribution by T2D status. B: Confusion matrix of the selected models. C: AUC-ROC curve of the selected mode.

migraine has a more pronounced impact on the accuracy of T2D detection in women than in men.

Lifestyle factors and symptoms also influence performance. The presence of depressive symptoms significantly impacts the algorithm's performance in women, increasing both specificity (0.75±0.05) and sensitivity (0.71±0.05). Conversely, for men, the impact of depressive symptoms are less prominent, with a slight decrease in sensitivity (from 0.75±0.05 to 0.71 ±0.06) yet a stable specificity of 0.71±0.07. This demonstrates enhanced accuracy in detecting T2D in women with depression. Smoking and stress revealed gender-specific impacts; smoking led to higher sensitivity in males (0.76±0.07) compared to a decreased sensitivity in females (0.53±12). Similarly, stress resulted in increased sensitivity for men (0.77±0.07) but decreased for women (0.62±0.07). Fatigue showed a uniform impact on specificity in both genders yet an increase in sensitivity in females with severe fatigue (0.68±0.05) compared to a stable sensitivity for males of (0.73±0.05).

Overall, the data indicates that the algorithm's specificity and sensitivity are influenced by demographic factors, comorbidities, and lifestyle factors, with notable differences observed between genders. These findings underscore the importance of considering these variables in the development and refinement of diagnostic tools, ensuring more accurate and gender-specific healthcare strategies in managing and diagnosing T2D.

**Table 3. Performance stratification of voice-based T2D status detection algorithms.**

| | | | Females | | | Males | | |
|---|---|---|---|---|---|---|---|---|
| | | | Specificity | Sensitivity | AUC | Specificity | Sensitivity | AUC |
| **Demographics** | **Age** | **<60 y** | 0.65 (0.04) | 0.65 (0.04) | 0.65 (0.03) | 0.70 (0.04) | 0.74 (0.04) | 0.72 (0.03) |
| | | **$\geq$ 60y** | 0.74 (0.12) | 0.74 (0.07) | 0.74 (0.07) | 0.70 (0.11) | 0.70 (0.10) | 0.70 (0.07) |
| | **Body Mass Index** | **<25 kg/m$^2$** | 0.68 (0.06) | 0.58 (0.12) | 0.63 (0.07) | 0.70 (0.06) | 0.78 (0.09) | 0.74 (0.05) |
| | | **$\geq$ 25 kg/m$^2$** | 0.65 (0.05) | 0.68 (0.04) | 0.67 (0.03) | 0.69 (0.05) | 0.72 (0.04) | 0.71 (0.03) |
| **Comorbidities** | **Hypertension** | **Present** | 0.76 (0.11) | 0.75 (0.05) | 0.75 (0.06) | 0.72 (0.11) | 0.76 (0.05) | 0.74 (0.06) |
| | | **Absent** | 0.65 (0.04) | 0.61 (0.05) | 0.63 (0.03) | 0.69 (0.04) | 0.70 (0.05) | 0.70 (0.03) |
| | **Migraine** | **Present** | 0.86 (0.07) | 0.75 (0.07) | 0.80 (0.05) | 0.67 (0.12) | 0.71 (0.11) | 0.69 (0.09) |
| | | **Absent** | 0.62 (0.04) | 0.65 (0.04) | 0.65 (0.04) | 0.70 (0.04) | 0.74 (0.04) | 0.72 (0.03) |
| **Lifestyle factors and symptoms** | **Smoking** | **Present** | 0.60 (0.09) | 0.53 (0.12) | 0.57 (0.07) | 0.74 (0.09) | 0.76 (0.07) | 0.75 (0.06) |
| | | **Absent** | 0.67 (0.04) | 0.69 (0.04) | 0.68 (0.03) | 0.69 (0.04) | 0.72 (0.04) | 0.71 (0.03) |
| | **Depressive symptoms** | **Severe** | 0.75 (0.05) | 0.71 (0.05) | 0.73 (0.03) | 0.71 (0.07) | 0.71 (0.06) | 0.71 (0.04) |
| | | **Mild** | 0.58 (0.05) | 0.61 (0.06) | 0.60 (0.04) | 0.69 (0.05) | 0.75 (0.05) | 0.72 (0.03) |
| | **Stress** | **Present** | 0.76 (0.07) | 0.62 (0.07) | 0.69 (0.05) | 0.69 (0.09) | 0.77 (0.07) | 0.72 (0.06) |
| | | **Absent** | 0.63 (0.04) | 0.70 (0.04) | 0.66 (0.03) | 0.70 (0.04) | 0.72 (0.04) | 0.71 (0.03) |
| | **Fatigue** | **Severe** | 0.68 (0.06) | 0.68 (0.05) | 0.68 (0.04) | 0.71 (0.06) | 0.73 (0.05) | 0.72 (0.04) |
| | | **Mild** | 0.65 (0.05) | 0.66 (0.06) | 0.65 (0.04) | 0.69 (0.05) | 0.73 (0.06) | 0.71 (0.04) |

## Agreement with ADA risk score

In the Bland-Altman analysis, the mean difference indicates the average bias between the algorithm's scores and the ADA risk scores. This analysis indicates that the algorithm has a mean difference of 0.57 for females and -0.15 for males compared to the ADA risk score, with over 93% agreement within acceptable limits for both genders, showing consistent agreement across genders (see S1 Fig).

Furthermore, we calculated the AUC for the ADA score and found comparable results to the voice-based algorithm's performance: AUC for the ADA risk score was 0.72 for females and 0.71 for males, compared to the algorithm's AUC of 0.71 (0.07) for females and 0.75 (0.05) for males. These findings indicate that our voice-based algorithm performs similarly to the established ADA risk score, further supporting its potential as a reliable screening tool for T2D.

## Discussion

In this study, using a large sample from the USA population, we developed voice-based algorithms to detect T2D status. Our goal was to explore the possibility of using a rapid, user-friendly voice recording as a T2D status predictor. We observed that the performance of the predictive algorithms was maximal when trained using the hybrid BYOL-S/CvT embeddings, achieving AUC scores of 0.75 and 0.71 for the male and female groups, respectively. Besides demonstrating overall fair to good performances, we also examined the influence of cofactors on voice-based T2D status prediction, which allowed us to identify key subgroups of the population with enhanced performances. In a sensitivity analysis, we have confirmed a strong agreement with the currently used questionnaire-based ADA risk score, a gold standard for T2D risk assessment in the USA.

Undiagnosed T2D or delayed diagnosis can accelerate the occurrence of serious diabetes-related complications, including cardiovascular diseases, neuropathy, retinopathy, and nephropathy [25]. One potential under-investigated effect of T2D is its impact on voice, which

may be due to the disease's influence on respiratory and neuromuscular functions [19,20,26]. It was already shown that pulmonary function is reduced in people with T2D compared to those with no diabetes [27]. For speech production, an individual needs a sufficient air intake, which then travels through the trachea and larynx, causing vocal fold vibrations. Articulating these vibrations into speech requires various small muscles in the neck and throat, connected by a large nerve network. Diabetes is commonly linked to peripheral neuropathy, but it can also impact other systems [28]. This includes potential nerve damage in the throat and neck region, which is vital for speech production. Research has suggested that diabetes can lead to voice changes, especially in those with poor glucose control, causing symptoms like hoarseness and strain [18,28]. These patients often have reduced maximum phonation times, indicating neuromuscular and respiratory alterations [18,20,28]. Building upon this, our study, with its larger sample size, offered a comprehensive exploration of the vocal and physiological complications associated with T2D. By assessing cofactors, we also highlighted how they influence voice patterns, providing valuable insights for future diagnostic strategies.

Key demographic indicators, mainly age, were central in T2D status prediction using voice, especially for women. This aligns with existing research that emphasizes the importance of this variable as a critical determinant of diabetes risk [29,30]. We observed that older females ($\geq$60) exhibited higher specificity, sensitivity, and AUC compared to younger ones (<60), but no difference was observed in males. An adult woman's average fundamental frequency range is 165 to 255 Hz, while a man's is 85 to 155 Hz [31]. In females, hormonal changes related to menopause can affect vocal cords and larynx and, consequently, cause a drop in the fundamental frequency of the voice [32]. These hormonal variations may interact with the metabolic disruptions caused by diabetes, leading to observable changes in voice pitch. On the other hand, males, not subject to the same degree of hormonal fluctuations, may exhibit less noticeable alterations in fundamental frequency.

Additionally, hypertension emerged as a key influencer in T2D status detection using voice, improving the predictive performance for both genders by up to 6%. While hypertension is known to be associated with diabetes development [33], it is not commonly incorporated into standard T2D risk assessment tools and its correlation with voice changes remains relatively unexplored [33,34].

Another distinguishing feature of our study is the gender-specific analysis of voice-based algorithms. We found that while certain determinants of T2D status were consistently influential across genders, others displayed gender-specific variations. Discrepancies observed in the impact of conditions such as migraines and on the voice-based T2D status detection algorithm's performance might be traced back to inherent gender-based physiological differences. Women are also more likely to experience migraine than men, with more frequent and severe attacks [35]. Besides, migraine and diabetes have already been shown to be associated with women [36], and our study confirms that this association can be captured by changes in female voices [37]. The varying impact of smoking on the algorithm's performances between genders may reflect gender-specific vocal changes caused by smoking [37,38]. Depression affects voices differently between men and women, suggesting that depression is linked to a higher risk of diabetes in women, but not in men [39,40]. This gender-specific association might explain the observed disparities in voice changes. The physiological and psychological stresses associated with depression may induce subtle voice changes that vary between genders, potentially due to hormonal or neurological differences. This variation might be more pronounced in women due to the combined impact of hormonal disruptions related to both diabetes and depression. Stress and fatigue, both of which can affect voice quality [41,42], seem to influence the algorithm's performance in a gender-specific manner. These factors, known to play roles in glucose

metabolism and insulin resistance [43], likely contribute to the voice patterns identified by the algorithm as indicative of T2D risk.

Such a sensitivity analysis is rarely performed in the field of vocal or digital biomarkers, as authors frequently report overall performances only. Our approach underscores that integrating the analysis of the influence of key demographic and health parameters is essential before developing any reliable voice-based screening tool. This helps to understand the potential physiological influence of these factors on either voice features or the health outcome of interest. Identifying the key sub-groups in the population is crucial to determining where the performance of these tools could be optimal.

## Strengths and limitations

This work has several strengths. First, we used the most comprehensive sample of USA participants with standardized voice ecological recordings, collected in a real-life setting, compared to existing datasets. Additionally, we performed the analysis separately, stratified for males and females, to account for major gender differences in voice characteristics and to mitigate gender bias. Voice features can vary significantly between males and females due to physiological and hormonal differences, which can affect the accuracy and performance of the algorithm if not accounted for. By developing separate models for each gender, we were able to fine-tune the algorithms for the specific characteristics of males and females, improving overall predictive performance and ensuring fairness and generalizability.

Besides displaying overall good performances, we also performed additional analyses to identify important subgroups where the voice-based algorithms would perform even better. Our comparative analysis of cofactors emphasized the complex nature of T2D and its interaction with voice characteristics, providing some levels of interpretability and explainability to the algorithms. Importantly, we have been able to benchmark the voice-based algorithms against an existing screening strategy in the USA, and we demonstrated a strong agreement with the ADA risk score. This concordance reinforces the potential use of voice-based analysis as a viable first-line screening tool for T2D.

There is also scope for further refinement before such algorithms can be considered ready for implementation as a screening tool and several limitations have to be acknowledged in our study. First, due to data constraints in ADA score calculation, missing values for parameters, namely physical activity and family history of diabetes were assigned a value of zero for all participants by default. While this approach might introduce less variability in the ADA scores, the potential for misclassification arises. However, the impact of this limitation is somewhat limited since the ADA score is primarily driven by age and BMI, which are available in our study. Even though they represent different constructs, we have still observed a strong agreement between the voice-based algorithms and the ADA risk score. Another limitation is that our study relied on a sample of English speakers only, with diverse T2D durations. To robustly establish and reinforce the performance of a future screening tool in predicting T2D, a more diverse and large dataset is needed, while specifically targeting early-stage T2D and prediabetes cases. Additionally, conducting longitudinal studies will help to better understand how changes in voice characteristics correlate with the development and progression of T2D. This approach will provide insights into the main clinical diabetes-related parameters, such as glycemic control and diabetes-related complications, and help establish causal relationships. Furthermore, it is also important to generalize this research across different populations, with diverse backgrounds and languages. Expanding datasets will allow a deeper examination of nuanced factors, comorbidities, and their interactions affecting voice-based screening tools in predicting T2D.

## Conclusion and perspectives

This work demonstrates the potential of using voice analysis in a diabetes context. A voice recording could potentially be soon used as a scalable, non-invasive first-line diabetes screening strategy. Future research should focus on targeting individuals with early-stage T2D and prediabetes and expanding our findings to other populations in prospective studies. Given the high societal costs of undiagnosed diabetes in the USA, our findings open new perspectives to improve secondary prevention, reduce the impact of diabetes and prevent severe complications and premature diabetes-related mortality.

## Supporting information

**S1 Fig. Bland-Altman plot** showing the agreement between the voice-based algorithms' predicted probability and the ADA risk score for both gender groups (A: Female group, B: Male group). Note: The predicted probability was scaled by a factor of 7 for harmonization. (TIF)

## Acknowledgments

We would like to thank all participants who contributed to the Colive Voice study, as well as our partners for their help in recruiting new participants. Special thanks go to Aurélie Fischer, Philippe Kayser, Luigi De Giovanni, Michael Schnell, and Aurore Dobosz for their substantial contribution to the Colive Voice study.

## Author Contributions

**Conceptualization:** Gloria Aguayo, Aurélie Fischer, Vladimir Despotovic, Guy Fagherazzi.

**Data curation:** Abir Elbéji, Mégane Pizzimenti, Hanin Ayadi, Vladimir Despotovic, Guy Fagherazzi.

**Formal analysis:** Abir Elbéji, Vladimir Despotovic.

**Methodology:** Abir Elbéji, Vladimir Despotovic, Guy Fagherazzi.

**Project administration:** Guy Fagherazzi.

**Resources:** Mégane Pizzimenti, Guy Fagherazzi.

**Supervision:** Guy Fagherazzi.

**Validation:** Abir Elbéji.

**Writing – original draft:** Abir Elbéji.

**Writing – review & editing:** Abir Elbéji, Mégane Pizzimenti, Gloria Aguayo, Aurélie Fischer, Hanin Ayadi, Franck Mauvais-Jarvis, Jean-Pierre Riveline, Vladimir Despotovic, Guy Fagherazzi.

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
