## [Decision Letter · Decision Letter 0]

5 Jul 2024

PDIG-D-24-00050

A voice-based algorithm can predict type 2 diabetes status in USA adults: Findings from the Colive Voice study

PLOS Digital Health

Dear Dr. Fagherazzi,

Thank you for submitting your manuscript to PLOS Digital Health. After careful consideration, we feel that it has merit but does not fully meet PLOS Digital Health's publication criteria as it currently stands. Therefore, we invite you to submit a revised version of the manuscript that addresses the points raised during the review process.

Please submit your revised manuscript within 60 days Sep 03 2024 11:59PM. If you will need more time than this to complete your revisions, please reply to this message or contact the journal office at digitalhealth@plos.org. Please include the following items when submitting your revised manuscript:

We look forward to receiving your revised manuscript.

Kind regards,

Ludwig Christian Giuseppe Hinske, M.D.

Academic Editor

PLOS Digital Health

Journal Requirements:

Additional Editor Comments (if provided):

Dear authors,

thank you for submitting your work at PLOS Digital Health. As you can see from the attached reviews, all reviewers found the manuscript interesting. However, major issues were identified, some of which could impact the value for the scientific community. As you can see from the reviews, many of the comments address definition criteria as well as the methodological approach, especially with respect to sample size and the risk of massive overfitting.

We would like to encourage you to work on these aspects in a revised version of your manuscript.

Sincerely,

 L. C. Hinske

Reviewers' comments:

Reviewer's Responses to Questions

**Comments to the Author**

1. Does this manuscript meet PLOS Digital Health’s publication criteria? Is the manuscript technically sound, and do the data support the conclusions? The manuscript must describe methodologically and ethically rigorous research with conclusions that are appropriately drawn based on the data presented.

Reviewer #1: Yes

Reviewer #2: Yes

Reviewer #3: Partly

2. Has the statistical analysis been performed appropriately and rigorously?

Reviewer #1: Yes

Reviewer #2: No

Reviewer #3: No

3. Have the authors made all data underlying the findings in their manuscript fully available (please refer to the Data Availability Statement at the start of the manuscript PDF file)?

Reviewer #1: Yes

Reviewer #2: Yes

Reviewer #3: Yes

4. Is the manuscript presented in an intelligible fashion and written in standard English?

Reviewer #1: Yes

Reviewer #2: Yes

Reviewer #3: Yes

5. Review Comments to the Author

Reviewer #1: The study explores using voice-based algorithms to predict T2D status in US adults, aiming to develop a non-invasive, scalable screening method. The authors analyzed text recordings from 607 Colive Voice study participants and used hybrid BYOL-S/CvT embeddings to create gender-specific algorithms for T2D prediction. The algorithms were evaluated using cross-validation, and their performance was stratified by age, BMI, hypertension, and compared to the ADA score for T2D risk assessment.

Comments

The study did not provide detailed information on the recruitment process and inclusion/exclusion criteria for participants. There may be potential selection bias if certain groups of individuals were more likely to participate in the study. 

The study included participants with diverse T2D durations, but did not specifically target early-stage T2D or prediabetes cases. 

Due to data constraints, physical activity levels and family history of diabetes were not available and were assigned a default value of zero for all participants. This may introduce less variability in the ADA scores and potential misclassification. 

While the study performed additional analyses to identify important subgroups and compared the influence of key demographic and health parameters, the interpretability and explainability of the algorithms could be further improved. 

The study did not account for potential confounding factors that may influence voice characteristics, such as smoking, alcohol consumption, or other underlying health conditions. 

The study used cross-sectional data, which limits the ability to establish causal relationships between voice characteristics and T2D status. 

The study relied on a sample of English speakers only, which may limit the generalizability of the findings to other languages and populations. 

The study did not include an external validation dataset to assess the performance of the developed algorithms. 

Although the study used a larger sample size compared to previous studies, the sample size may still be insufficient to capture the full spectrum of voice variations associated with T2D. 

While the study compared the performance of the voice-based algorithms with the ADA risk score, it did not compare them with other established screening methods, such as fasting blood glucose or HbA1c tests.

Reviewer #2: The manuscript by Abir Elbeji et al developed a novel screening tool for diagnosing type 2 diabetes mellitus by building a voice-based machine-learning algorithm.

Although the data presented is clearly interesting, there are several issues that will require further clarification prior to publication:

1. The authors are advised to clearly describe how type 2 DM are diagnosed and defined. What is the diagnositc criteria for Type2 DM in this study?

2. The authors are requested to calculate the AUC by diagnosisng/evaluating participants with ADA risk scores.

3. The authors are also requested to compare the AUC above with that of the develpped algorithm.

4. What does the numerical value in the brackets in Table 2 mean? Are they standard deviation? or standar error mean?

5. In line 297, the authors wrote, "notable differences were observed for females across...". In addition, they also wrote in line 305 that no noticeable disparities were observed among males. The authors are requested to clarify the difference between "notable difference in women" and "no noticeable disparities among males". The authors are advised to describe how they defined "notable/noticeable difference".

6. In line 315, the authors wrote that the presence of depression significantly influenced the algorithm's performance in woman. The authors need to show the evidence of this significance.

7. In line 344, they wrote AUC score of algorithm in female T2DM as 0.72. However, I'm afraid that this might be 0.71 (I think this is a innocent mistake/mistyping)

Reviewer #3: The present manuscript highlights an algorithm to detect type 2 diabetes (t2d) based on multidimensional data features derived from voice recordings. I have a few comments/questions for the authors. Thank you for considering my comments.

1. Introduction: The authors say the FINDRISC (Finnish diabetes risk score) has limited detection capabilities (AUC of 76%). However, their algorithm's AUC is similar or lower (75% for males, 71 for females). Would this not be an argument to use a much simpler questionnaire to assess the risk for t2b?

2. Sample size: The overall sample size is N=607, reported in the abstract. However, the authors performed the analyses stratified by gender. The larger group of females is N=323, with 162 events (t2d) and 161 non-events. Thus, the effective sample size for their model is only N=161. This is a rather small study to develop a model based on 200 features after dimensionality reduction. The number of features is higher as the number of events observed; how do the authors mitigate massive overfitting?

3. Methods: The authors state they use TRIPOD reporting guidelines; however, it was not reported whether the study had missing data or not and how missing data was handled if present.

4. Methods: What was the rationale for performing the analysis separately, stratified for males and females? Why was the model not developed on all the data, and why is sex used as one of the prognostic factors along the features?

5. Methods: How was the number of components in the PCA determined?

6. Methods: I was wondering about the performance of a very simple logistic model using sex, age, hypertension, and BMI as diagnostic factors. Would that be feasible as a benchmark?

7. Discussion/Conclusion: The authors suggest the tool as a screening strategy for t2d; however, what is the optimal threshold to be used? For clinical implementation, this would require a decision curve analysis. Maybe the author could discuss this point.

6. PLOS authors have the option to publish the peer review history of their article (what does this mean?). If published, this will include your full peer review and any attached files.

**Do you want your identity to be public for this peer review?** For information about this choice, including consent withdrawal, please see our Privacy Policy.

Reviewer #1: No

Reviewer #2: Yes: Satoru Tada

Reviewer #3: No

---

## [Decision Letter · Decision Letter 1]

23 Oct 2024

A voice-based algorithm can predict type 2 diabetes status in USA adults: Findings from the Colive Voice study

PDIG-D-24-00050R1

Dear Dr. Fagherazzi,

We are pleased to inform you that your manuscript 'A voice-based algorithm can predict type 2 diabetes status in USA adults: Findings from the Colive Voice study' has been provisionally accepted for publication in PLOS Digital Health. We apologize for the long processing time, since we were hoping to get feedback from all three reviewers of the initial submission. Thanking you for reaching out to us, explaining the situation regarding the time constraints.

Best regards,

Ludwig Christian Giuseppe Hinske, M.D.

Academic Editor

PLOS Digital Health

Reviewer Comments (if any, and for reference):

Reviewer's Responses to Questions

**Comments to the Author**

1. If the authors have adequately addressed your comments raised in a previous round of review and you feel that this manuscript is now acceptable for publication, you may indicate that here to bypass the “Comments to the Author” section, enter your conflict of interest statement in the “Confidential to Editor” section, and submit your "Accept" recommendation.

Reviewer #1: All comments have been addressed

Reviewer #2: All comments have been addressed

2. Does this manuscript meet PLOS Digital Health’s publication criteria? Is the manuscript technically sound, and do the data support the conclusions? The manuscript must describe methodologically and ethically rigorous research with conclusions that are appropriately drawn based on the data presented.

Reviewer #1: Yes

Reviewer #2: Yes

3. Has the statistical analysis been performed appropriately and rigorously?

Reviewer #1: Yes

Reviewer #2: Yes

4. Have the authors made all data underlying the findings in their manuscript fully available (please refer to the Data Availability Statement at the start of the manuscript PDF file)?

Reviewer #1: Yes

Reviewer #2: Yes

5. Is the manuscript presented in an intelligible fashion and written in standard English?

Reviewer #1: Yes

Reviewer #2: Yes

6. Review Comments to the Author

Reviewer #1: The revised manuscript shows significant improvement in terms of clarity, context, and scientific rigor. The additions, particularly the ADA risk score comparison, strengthen the paper's contribution to the field.

Reviewer #2: The authors have addressed my concerns.

I am pleased to inform you that this manuscript has now been accepted for publication.

7. PLOS authors have the option to publish the peer review history of their article (what does this mean?). If published, this will include your full peer review and any attached files.

**Do you want your identity to be public for this peer review?** For information about this choice, including consent withdrawal, please see our Privacy Policy.

Reviewer #1: No

Reviewer #2: **Yes: **Satoru Tada, MD, PhD
